# *Pleurotus* Genus as a Potential Ingredient for Meat Products

**DOI:** 10.3390/foods11060779

**Published:** 2022-03-08

**Authors:** Brisa del Mar Torres-Martínez, Rey David Vargas-Sánchez, Gastón Ramón Torrescano-Urrutia, Martin Esqueda, Javier Germán Rodríguez-Carpena, Juana Fernández-López, Jose Angel Perez-Alvarez, Armida Sánchez-Escalante

**Affiliations:** 1Coordinación de Tecnología de Alimentos de Origen Animal (CTAOA), Centro de Investigación en Alimentación y Desarrollo, A.C. (CIAD), Carretera Gustavo Enrique Astiazarán Rosas 46, Hermosillo 83304, Mexico; brisa.torres@estudiantes.ciad.mx (B.d.M.T.-M.); rey.vargas@ciad.mx (R.D.V.-S.); gtorrescano@ciad.mx (G.R.T.-U.); esqueda@ciad.mx (M.E.); 2Consejo Nacional de Ciencia y Tecnología, Av. Insurgentes Sur, 1582, México City 03940, Mexico; 3Unidad Académica de Medicina Veterinaria y Zootecnia, Universidad Autónoma de Nayarit, Compostela 67300, Mexico; german.rc@uan.edu.mx; 4IPOA Research Group, Centro de Investigación e Innovación Agroalimentaria y Agroambiental, Miguel Hernández University (CIAGRO-UMH), Orihuela, 03312 Alicante, Spain; j.fernandez@umh.es (J.F.-L.); ja.perez@umh.es (J.A.P.-A.)

**Keywords:** edible mushrooms, chemical composition, bioactive compounds, meat products

## Abstract

Edible mushrooms are considered an important source of nutritional and bioactive compounds. In this review, the findings of macronutrients, bioactive compounds, antioxidant activity, and antimicrobials against foodborne pathogens of some *Pleurotus* spp., as well as their potential use as an ingredient in the meat industry are discussed. The results show that *Pleurotus* spp. are an important source of proteins and amino acids, carbohydrates, minerals, and vitamins. Additionally, the presence of some bioactive components, such as polysaccharides (α-glucans, β-glucans, and so on), proteins/enzymes and peptides (eryngin, pleurostrin, and others) phenolic acids (*p*-coumaric, chlorogenic, cinnamic, ferulic, gallic, protocatechuic, and others) and flavonoids (chrysin, naringenin, myricetin, quercetin, rutin, or the like) has been demonstrated. Several works evidenced the use of *Pleurotus* spp. in some meat and meat products (patties, sausages, paste, and suchlike) as a novel ingredient in order to improve their chemical composition and functional health promoting properties, as well as to increase their physicochemical and sensory attributes. In conclusion, the use of *Pleurotus* is a promissory strategy for the development of natural additives rich in nutritional and bioactive components for meat and meat product formulation.

## 1. Introduction

One of the oldest food products with a record of their use are edible mushrooms, which are recognized for their nutritional and culinary value, thanks to their unique taste and texture. Furthermore, in recent decades, their potential biological activity has been proven; this is attributed to the presence of secondary metabolites. Consequently, in the presence of these, fungi show antioxidant, hypocholesterolemic, antimicrobial, immunomodulatory activity, among others. Thus, mushrooms can be considered functional foods since they can provide health benefits beyond the nutrients that they contain [1].

According to their usefulness, mushrooms can be divided into four categories: edible, medicinal, poisonous mushrooms, and finally, those whose properties remain less defined, including a significant number of these. Furthermore, they can be classified into various ecological groups, of which the most important are the saprophytes; Other classifications are based on growth with soil, mycorrhizal, lignicolous, hallucinogens, and coprophilous [2,3].

Edible mushrooms can be defined as “macro fungus with a distinctive fruiting body, which can be epigeal (above ground) or hypogeal (below ground) and large enough to be seen with the naked eye and picked by hand” [1]. The fruiting bodies are found mainly on the ground, varying in size, shape, and coloration for each species. These have filamentous bodies delimited by cell walls, are not mobile, and reproduce sexually and asexually by spores. They grow from spacious mycelia (hyphae), mainly underground, by the fruiting process. The mycelium is the vegetative part of a fungus. It consists of a system of branching hyphae through the soil, compost or substrate, the wood trunk, or other lignocellulosic material. After a period of growth and under favorable conditions, the established (matured) mycelium produces a fruitful structure called “fungus” [4,5].

Nowadays, mushrooms are used in medicine, pharmacy, food, and fermentation fields; they are considered a rich source of protein because they contain all essential amino acids, plus fiber and little fat. Likewise, they provide significant amounts of vitamins and bioactive compounds, such as unsaturated fatty acids, phenolic compounds, tocopherols, and carotenoids [6].

Many countries consume mushrooms as a delicacy, particularly for their aroma and texture, fiber content, in addition to their low energy intake. According to the rules of mycological nomenclature, there are around 120,000 species of fungi, which represent 3–8% of the estimated number of species existing on Earth. More than 2000 species are safe for consumption, and approximately 700 species are known to possess significant pharmacological properties [5,7]. Technological improvements have made its cultivation possible all over the world; the production of mushrooms on a global scale is represented by about 85% by five main species or genera: *Agaricus bisporus* (estimated at 30% of the world mushroom production), the genus *Pleurotus* (from five to six species cultivated in approximately 27%), *Lentinula edodes* (17%), *Auricularia* (6%), and *Flammulina* (5%) [8].

*Pleurotus ostreatus* is one of the most relevant edible fungi at the production level, a saprophytic fungus characterized by having bioactive compounds, mainly of phenolic origin. In the last decades, its utilization has grown due to the potential beneficial effects that its consumption can contribute to health. Even though the demand for mushrooms has been outstanding: growing to satisfy it, the yield is rarely doubled in today’s agriculture [9].

Moreover, meat and meat products are essential for the human diet due to their nutritional components, including proteins, fats, minerals, vitamins, etc. However, some of these components are associated with human health risks, as well as related to loss of quality [10]. In this way, the meat industry modified meat products by incorporating natural additives (rosemary, grape, avocado, green tea, cocoa, broccoli, edible mushroom, among others) in order to replace nutritional components, increase functional-health promoting components, and enhance biological activity [10,11,12].

Therefore, the present review summarizes relevant studies about the composition, bioactive properties, and uses of *Pleurotus* spp. as a natural food additive for meat and meat products.

## 2. *Pleurotus* spp.

*Pleurotus* spp., are edible mushrooms from the *Basidiomycetes* that belong to the *Pleurotaceae* family (*Agaricales*, *Agaricomycetes*), commonly known as “oyster mushrooms” because of the form of their fruiting bodies. These species are the second most important in the commercial context. These mushrooms are popular due to their texture, aroma, and taste besides the possible effect on human health for the bioactive compounds (polysaccharides, β-glucans, considered as proven functional food [13,14], proteins/enzymes, peptides, lectins, terpenoids, polyketides, and phenolic compounds) it has which could be applied as medicines or to the human diet. This genus comprises about 200 spices, which are distributed worldwide, and can be found in ecological niches, more preferably rotten tree trunks and branches but can be grown in a wide range of temperate and tropical climates; oyster mushroom preferentially decays lignin instead of polysaccharides. For this reason, they are known as white rot fungi [15,16,17,18]. The species have carpophores with eccentric pileus and decurrent blades that show a white or hyaline color enhanced by cylindrical or oval shapes like an oyster shell [19].

The *Pleurotus* genus is among the species with the highest production worldwide. The species of fungi of this genus exhibit multidirectional effects that promote health. Additionally, it is called the poor man’s meat this is because a variety of substrates considered waste are used to grow mushrooms in rural conditions, which makes mushroom proteins more affordable and more available than proteins of animal origin. Some authors suggest that edible mushrooms can be classified as a functional food due to the possible positive effect that their consumption causes on the human body [20,21,22,23].

These mushrooms are capable of colonizing and degrading many lignocellulosic residues (cotton waste, walnut shell, straws of rice, wheat, sorghum, maize, wood), and they are a source of important nutrients and healing properties. Compared to other types of mushrooms, these require less growth time and are relatively rarely attacked by diseases and pests [24,25].

Within the genus *Pleurotus,* the most common species are grey abalone oyster (*P. sajor-caju),* recently classified on *Lentinus* genera, pink oyster mushroom *(P. djamor*), king oyster mushroom (*P. eryngii)*, golden oyster (*P. citrinopileatus*), branched oyster mushroom (*P. cornucopiae*), king tuber oyster mushroom (*P. tuber-regium*), phoenix oyster (*P. pulmonarius)*, abalone mushroom (*P. cystidiosus)*, white ferula mushroom (*P. nebrodensis*), and oyster mushroom (*P. ostreatus*; Figure 1) [15].

## 3. Macro and Micronutrients

The approximate chemical composition of *Pleurotus* spp. (Table 1) has been reported in previous works. Its fruiting body and mycelial biomass composition are determined by a range of characteristic aromas and flavors that are due to carbohydrates, lactones, amino acids, and terpenes [26]. These include high protein content, and essential amino acids enabling it to be as a substitute for meat diet; a chitin rich wall acts as a source of dietary fiber, vitamin content (B_1_, B_2_, B_12_, C, D, and E), micro and macro-elements, carbohydrates, low fat content, and almost zero cholesterol content. In relation to its composition of chemical structures, such as secondary metabolites, such as betalains and alkaloids, as well as glycoproteins and polysaccharides, *Pleurotus* genus is one of the most diverse edible and medicinal mushrooms [27].

Concerning their chemical composition, the mushrooms cultivated commercially are characterized by showing a similar profile of nutritional components compared to wild mushrooms. However, there may be certain differences related to their qualitative and quantitative composition, which will depend on factors, such as the strain of the fungus, the extraction process of the chemical compounds from the morphological parts of the fungus, and to a large extent on the conditions adopted during the cultivation process [33].

*P. ostreatus* can be considered an important source of proteins (7.3–53.3%); compared with other foods, oyster mushrooms have all nine essential amino acids, for this reason, are ranked below animal meats and can be considered as a substitute for a meat diet. However, the amount of protein can be variable it has been reported that many factors affect it, such as the stage of maturation, the type of mushroom, the harvest location, and availability of nitrogen content in the medium; therefore, if a substrate rich in nitrogen or supplemented with nitrogen sources is used in the cultivation of it, the protein content could be higher [16,34,35,36].

*Pleurotus* spp. has an important amount of carbohydrates (50–60% dry weight); these consist principally of sugars (oligosaccharides, monosaccharides, and disaccharides) which are correlated to the synthesis of polysaccharides. Generally, these are represented by fiber, such as glycoproteins that include chitin, α-β, and glucans, cellulose, besides other hemicelluloses, such as mannans, galactans, and xylans [16,36,37].

About minerals, in general, *Pleurotus* species are an important source, contain a mix of nutritionally essential minerals, such as potassium (933–967 mg/100 g), notable levels of phosphorus (212–224 mg/100 g), calcium (221–238 mg/100 g), magnesium (366–407 mg/100 g), and in minor level sodium (40–46 mg/100 g), these represents the majority of fruit bodies, other minerals include copper (465–732 µg/g), zinc (113–131 mg/100 g), iron (105–112 µg/g), and cadmium (5–35 µg/g), which represents minor components [16,37,38,39,40].

Regarding micronutrients, Pleurotus could be an important source of vitamins, such as niacin, riboflavin, and folates, as good as vegetables, but the bioavailability of the latter is better for folic acid in contrast to peas or spinach. Furthermore, mushrooms contain a minor amount of vitamin C (12–15 mg/100 g) and thiamine (B_1_) (0.2–0.4 mg/100 g), niacin (vitamin B_5_) (6–9 mg/100 g), pyridoxine (vitamin B_6_) (0.1 mg/100 g), retinol (vitamin A) (25–26 μg/100 g), riboflavin (B_2_) (0.5–0.7 mg/100 g), and ergocalciferol (D_2_) (0.8–0.9 µg/g) among others [16,37,41,42,43,44,45].

This genus, characterized by their low-fat content, and therefore, are classified as low calorific food, with a small quantity of cholesterol and fat, lower than 4%. However, as mentioned above, this amount depends on the substrate used in the cultivation. It has been reported that the most dominant unsaturated fatty acid in mushrooms is linolenic acid (C18:2), an essential fatty acid, which is essential for the production and conversion of flavor components in *Pleurotus*. Besides, other fatty acids reported are oleic (C18:1) and palmitic (C16:0) [16,35,42,46].

## 4. Bioactive Compounds and Their Activity

Plant material composition includes a wide content of physiologically active compounds, which exert beneficial effects on human health and can reduce the risk of some chronic diseases. In addition, the human health benefits and the functional potential of these active compounds in the food industry have been demonstrated. In this context, mushrooms are becoming increasingly popular as a source of nutritional, nutraceutical, and functional components [1,15]. Previously, the presence of some bioactive components of *Pleurotus* spp. has been evidenced. The Bioactive compounds in *Pleurotus* mushrooms include high and low molecular weight. Polysaccharides (β-glucans), proteins/enzymes, and peptides are included in the high molecular compounds division, as for the low side, fatty acids, esters, terpenes, and polyphenols can be found [20]. For example, it has been reported that *P. ostreatus* extracts have antiviral, antiproliferative, anti-inflammatory activity, among others, which are associated with dietary fiber and other polysaccharides’ biological activity [15,34,47]. Additionally, the anti-inflammatory effects of *Pleurotus* spp. have been demonstrated when using in vivo models against dermatitis and arthritis [47]. While in another study, it was observed that supplementation of β-glucans (1 mg/kg) isolated from *P. ostreatus* decreased swelling and arthritic scores in rats [48]. One of the primary antioxidants is the phenolic compounds; these are mainly free radical scavengers that prolong or delay the beginning of lipid oxidation, decreasing the formation of the volatile decomposition products, which could cause rancidity [49]. Regarding phenolic compounds, *Pleurotus* spp. contain several types (Table 2), including phenolic acid whose bioactivity and antioxidant properties are associated with the ring structures attached with the phenolic hydroxyl groups, and flavonoids, which are extensively associated with their human health benefits and with its functional potential in food matrices [15,50,51].

### 4.1. Antioxidant Activity

Diverse stress factors could disrupt normal cellular functions and initiate chain reactions that compromise the integrity of cells, consequently producing high amounts of reactive oxygen species. The generation of these metabolites can damage cellular structures (oxidation of lipids and proteins). One of the most controversial topics in chemistry is the oxidation of lipids and proteins in food [58,59]. Furthermore, deficiency in endogenous antioxidant defense can result in oxidative stress associated with various health problems, including coronary heart disease, among others [60]. Therefore, in various research works, the antioxidant effect of extracts obtained from *P. ostreatus* against the formation of radicals that affect human health has been evaluated [61]. Jayakumar et al. [41] assessed the potential of an ethanolic extract of *P. ostreatus* to inhibit the damage of deoxyribose mediated by hydroxyl radicals, finding that the eliminating effect of the hydroxyl radical of the fungus extract, at a concentration of 10 mg/mL, was of 56%. The antioxidant compounds found in the fruiting bodies, the mycelium, and the mushroom culture can be phenolic acids, polysaccharides, tocopherols, flavonoids, carotenoids, glycosides, ergothioneine, and ascorbic acid [8]. Although the antioxidant activity of *Pleurotus* spp., also has been associated with some proteins/enzymes and peptides [62].

Acosta-Urdapilleta et al. [44] reported the inhibition of ABTS^•+^ and DPPH^•^ radicals from 96% to 53% and 98% to 24%, respectively, in the evaluation of antioxidant activity; *P. citrinopileatus*, showed the highest antioxidant activity. In another study, the assessment of antioxidant activities, and bioactive compounds of *P. sajor-caju* was evaluated. High inhibition (>50%) of the DPPH^•^ radical with methanolic extract of the *P. sajor-caju* was recorded (67%), which could be associated with the presence of flavonoids, alkaloids, and phenolic bioactive metabolites [63]. The antioxidant activity of *Pleurotus ostreatus* has been reported; methanolic extracts showed inhibition of the DPPH^•^ radical from 47 to up 50% in a study by Stefan et al. [64]; also report activity of 20% of inhibition of ABTS^•+^ radical and 74 to 79% of chelating activity. Ghosh et al. [65] compared the antioxidant properties and made a phytochemical screening of extracts from three cultivated *Pleurotus* species, *P. eous, P. florida, and P. ostreatus.* In their study results reported a high amount of phenols (13.03 μg gallic acid equivalent/mg of extract), flavonoids (3.57 μg quercetin equivalent/mg of extract), and ascorbic acid (16.66 μg/mg of extract) for the *P. eous* extracts; furthermore, potential free radical scavenging, the extract showed inhibition >50% of the DPPH^•^ radical (*P. florida* 64%, *P. ostreatus* 67% and *P. eous* 72%); while the inhibition of the ABTS^•+^ radical showed scavenging from 20% to 78%. In another report, Fasoranti et al. [66], exhibit the antioxidant activity of *P. pulmonarius and P. ostreatus*, cultivated on substrate fortified with selenium, and the results showed a range of 27% to 97% of DPPH^•^ scavenging effect at concentrations of 50 and 250 μg/mL, respectively, whereas 69% to 91% of hydroxyl ion scavenging ability.

### 4.2. Antimicrobial Activity

Another property of great interest is its antimicrobial activity, which has been extensively evaluated in vitro. Shen et al. [67], reported inhibition of pathogenic bacteria, both Gram-positive (*Bacillus, Clostridium perfringens, Staphylococcus* spp.) and Gram-negative (*Escherichia coli*, *Klebsiella pneumonia, Pseudomonas aeuroginosa, Salmonella* spp., *Shigella* spp.), by extracts of *P. ostreatus* [67,68]. Akyuz and Kibag [69] evaluated the antimicrobial activity of *P. eryngii* cultivated on two different substrates, using the disk diffusion method. In the study, they reported that its extracts inhibited the growth of microorganisms (7.7–10.3 mm), *E. coli, S. aureus*, among others. The antimicrobial activity of *P. plulmonarius* was evaluated by Adebayo et al. [70] using the agar well diffusion technique. The results show the major inhibition zone was shown for *S. aureus* (30 mm), while the smallest zone size was obtained against *E. coli* (7 mm). Additionally, the antimicrobial potential of *P. ostreatus* fruiting body and mycelium extracts against *Candida albicans*, *E. coli,* and *B. subtilis* has been evidenced [71].

Extracts of fruiting body powder from *P. ostreatus*, showed antimicrobial activity against *C. albicans*; the result of the assay showed the formation of barriers on the extract concentration zone of 10.8 mm width [72]. A comparative study of *Pleurotus* spp. realized by Sathyan et al. [73], showed antimicrobial activity against *E. coli*, *P. aeruginosa* and *K. pneumonia* from *P. ostreatus, P. djamor,* and *P. enryngii,* respectively. Likewise, Adebayo et al. [74] evaluated antibacterial properties of hydroalcoholic extracts from four species of *Pleurotus* following the minimal inhibitory concentration and the minimal bactericidal concentration assays, finding that most bacteria were susceptible to a minimal inhibitory concentration in a range from 11 to 100 µg/mL of mushroom extracts, while only two of the *Pleurotus* spp. extracts showed bactericidal activity; they showed a bactericidal effect to *Stenotrophomonas* spp. from *P. ostreatus*; while *P. tuber-regium* showed a bactericidal effect for *Streptococcus agalactiae, E. coli, B. subtilis,* and *P. aeruginosa*. In another study, Pandey et al. [75] showed the antimicrobial activity of methanol extracts of *P. flabellatus*; the highest activity was reported against *Proteus mirabilis,* these results could be attributed to the presence of secondary metabolites, such as amino acids, alkaloids, flavonoids, etc. Although the antibacterial activity of proteins/enzymes has also been demonstrated, as well as peptides from *Pleurotus* spp. For example, the antibacterial properties of *P*. *eryngii* and *P. ostreatus* have been investigated against *Bacillus* sp. and associated with the presence of eryngeolysin and ostreolysin proteins, respectively [76].

### 4.3. Biological Activity

*P. ostreatus* also has shown other interesting healthy properties. As mentioned above, fungi have an important source of carbohydrates (polysaccharides) in their cell walls; a beneficial effect on health has been reported, and this activity is mainly related to its immunomodulatory and anticancer properties. These polysaccharides induce the immune system through dectin-1, CR3, and lactosylceramide sequestering receptors, which are involved in macrophage activation, phagocytosis, and cytokine production. Likewise, they can activate neutrophil complement receptors [15]. Studies conducted by Radzki et al. [47] isolated water-soluble polysaccharides (WSP) from *P. ostreatus* fruiting body and tested their antiproliferative activity against MCF-7 and T-47D breast cancer cell lines; the results showed that the most evident effect was obtained when using a concentration of 250 mg/mL, obtaining cell viability of 65 and 72%, respectively.

Another important biological property of this fungus is its antihypercholesterolemic activity. Research by Alam et al. [77], evaluated the effect of the inclusion of 5% of *P. ostreatus flour* in the diet of hypercholesterolemic rats on liver and kidney functions. The results obtained in this study showed a decrease of 21% and 45% in the level of total cholesterol in plasma and triglycerides due to the diet added with *P. ostreatus*, respectively.

Thus, Corrêa et al. [34] mentioned that this mushroom has antinociceptives, hypoglycemic and hypolipidemic effects, in addition to other interesting effects, such as cytoprotectors, prevention of skin aging, neurogenic and anticataratogenic.

## 5. *Pleurotus* spp. as an Ingredient in Meat Products

Food additives are substances deliberately added to food in small amounts along production or processing to enhance its organoleptic characteristics, besides, to delay deterioration during storage and protect the consumer against food poisoning [78]. According to the Food Protection Committee of the Food and Nutrition Board, food additives are defined as ingredients or mixtures of different ingredients, other than a basic food product, which are present in food because of any aspect of storage, processing, production, or packaging. Additives can be classified into six main categories: nutritional additives, texturizers, preservatives, miscellaneous additives, flavorings, and colorants [79].

In this context, during meat product processing, synthetic and natural chemical compounds (GRAS, generally recognized as safe) are used to ensure the safety and quality of the final product. It is well known that raw muscle and fat are the principal ingredients in a meat product. While, food additives, including non-meat ingredients, such as small molecules (phosphate, salt, antimicrobial, and antioxidant compounds that can be found in plants, spices, among others) or bigger molecules (proteins, gums, starch, non-muscle, among others), are also used to enhance meat and meat product quality [80].

Lipid and protein oxidation, along with microbial contamination, are the main cause of spoilage in meat and meat products and are closely associated with the appearance of undesirable changes in color, taste, texture, and nutritional value. To maintain quality standards, the meat industry has used natural antioxidants to reduce these problems. Previous studies have shown that plant materials, such as edible mushrooms are extensively investigated due to the potential they may have as a food ingredient [10,81].

Moreover, to ensure the safety of an edible mushroom when used as a food ingredient, it is important to know its microbial load and identify its allergenic components. In previous work, the microbial load and the occurrence of foodborne pathogens of fresh and stored edible mushroom (*P. ostreatus* and *P. eryngii*) was investigated. The results revealed that samples showed low contamination (aerobic mesophilic bacteria <5.0 log cfu/g), however, the microbiological quality was reduced during mushroom storage (4 °C for 12 days). Regarding identified foodborne pathogens, *Pseudomonas fluorescens* and *Ewingella americana* were the most abundant bacteria. In addition, *Salmonella* spp., *L*. *monocytogenes* and *Bacillus cereus* were absent in the tested samples [82]. In addition, it has been reported that *Pleurotus* spp. spores are associated with some allergy risks, such as asthma and acute IgE-mediate rhinoconjunctivitis in sensitized individuals, therefore, research has been carried out to reduce the content of spores through mutation processes [83].

The technological properties that *Pleurotus* spp. could provide to meat products have been reported; for example, Stephan et al. [84] evaluated the mycelia of *P. sapidus* as a protein source in a vegan boiled sausage and compared it to commercial proteins and meat sausages. The results showed enhancement in the texture profile of the vegan sausage and similar flavor after production.

Considering the health benefits and functional properties of edible mushrooms, *Pleurotus* can be proposed as a potential source of natural additives to improve meat and meat products’ quality (Table 3).

*Pleurotus* spp. has been used in meat products to improve meat products. Table 3 shows the effects of the addition of *Pleurotus* spp. in different meat products. In this table, it can be observed that the most addressed aspects evaluated in these works are the effects of *Pleurotus* over the technological properties of these meat products, not only on fresh products also on cooked ones. Furthermore, another aspect that has been reported is the antioxidant effect that the addition of *Pleurotus* could have over the meat products; these can be associated with the bioactive compounds found in the edible mushroom. Besides the antioxidant effects, the flavor enhancement in the formulations and the sensory attributes have been reported too.

Despite the great benefits that the consumption of this mushroom brings, both to health and the potential use as a meat product ingredient, there is not much evidence about its use. Which opens an area of opportunity for the elaboration of products added with these mushrooms, as well as the study of the benefits that their consumption can provide to health.

## 6. Conclusions

Edible mushrooms, including some *Pleurotus* spp., have long been considered one of the most delicious and succulent foods worldwide. Furthermore, several studies demonstrated the presence of nutritional (proteins and amino acids, fat and fatty acids, vitamins, etc.) and biologically active components (i.e., polysaccharides, proteins/enzymes, peptides, phenolic compounds, among others), which are associated with their antioxidant and antimicrobial potential. On the other hand, it has been demonstrated the use of these mushroom species is an opportunity to improve, the nutritional and functional-health properties of meat (beef, chicken, and pork) and meat products (patties, nuggets, sausages, meatballs, Salami, paste, and others) as well as to enhance their quality.

## Figures and Tables

**Figure 1 foods-11-00779-f001:**
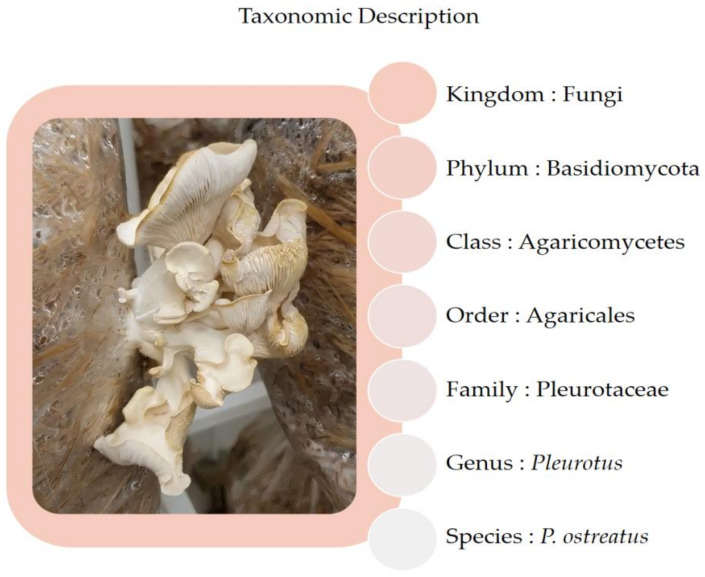
Taxonomic description and fruit bodies of *Pleurotus ostreatus*.

**Table 1 foods-11-00779-t001:** Proximate composition (% dry matter) of some *Pleurotus* species.

Species	Protein	Fat	Ash	Carbohydrates	Reference
*P. ostreatus*	7	1.4	5.7	85.9	[28]
*P. eryngii*	11.0	1.5	6.2	81.4	[29]
*P. pulmonarius*	23.2	4.2	4.8	50.1	[30]
*P. sajor-caju*	26.34	3.67	10.37	38.17	[31]
*P. florida*	20.56	4.3	9.02	42.83	[32]

**Table 2 foods-11-00779-t002:** Phenolic components identified in some *Pleurotus* spp.

Specie	Compound	References
*P*. *ostreatus*	Phenolic acids: gallic, protocatechuic, cinnamic, *p*-coumaric, ferulic Flavonoids: chrysin	[52]
	Phenolic acids: chlorogenic, syringic, ferulic, cinnamic, *p*-coumaric, caffeic, vanillic Flavonoids: naringenin	[53]
*P. pulmonarius*	Phenolic acids: gallic, homonogentisic, protocatechuic, chlorogenic, vanillic Flavonoids: (+)-catechin, naringin, myricetin, resveratrol, quercetin	[54]
	Phenolic acids: gallic, caffeic, vanillic, *p*-coumaric, ferulic	[55]
*P. eryngii*	Phenolic acids: 2,4-dihydroxybenzoic, chlorogenic, syringic, ferulic, cinnamic, *p*-coumaric, caffeic, vanillic Flavonoids: naringenin	[53]
	Phenolic acids: chlorogenic, *p*-hydroxybenzoic, *p*-anisic, ferulic, sinapic, syringic, vanillic Flavonoids: catechin, epicatechin, rutin, myricetin, hesperidin, quercetin	[56]
	Flavonoids: rutin	[57]
*P. sajor-caju*	Phenolic acids: cinnamic	[51]

**Table 3 foods-11-00779-t003:** *Pleurotus* spp. enhanced meat products quality.

Meat Product	Mushroom	Relevant Results	Reference
Chicken patties	*P. sajor-caju*/fresh ground	Decreased lightness, yellowness, hardness, and chewiness while increase springiness	[85]
	*P. sajor-caju*/fresh ground	Low fat content without threatening sensorial properties	[86]
	*P. sapidus*/flour	Highest antioxidant activity	[87]
	*P. sajor-caju*/fresh ground	Reduction of fat, increased the ash content, no difference recorded in sensory attributes	[88]
	*P. sajor-caju*/fresh ground	Lower costs of production, enhancement of nutritional composition	[89]
Chicken nugget	*Pleurotus sajor-caju*/flour	Higher lightness, chewiness, springiness, water activity, and moisture	[90]
Chicken sausage	*Pleurotus sajor-caju*/fresh ground	Increased moisture and fiber but reduced the ash content, crude fat, and crude protein	[91]
	*Pleurotus ostreatus* and *P. nebrodensis*/Blanched	The addition improves the taste and texture	[92]
	*P. sajor-caju*/flour	Higher nutritional composition: increment total dietary fiber, low fat content, and β-glucan except for protein	[93]
Beef brain sausage	*P. ostreatus*/blanched paste	50:50 treatment presents the best characteristics and preference in organoleptic test	[94]
Beef patties	*P. eryngii*/flour	Lowest reduction in diameter, thickness, and weight loss duringcooking, besides improved flavor, juiciness, tenderness, and acceptance.	[95]
	*P. ostreatus*/flour	Reduce fat and sodium content besides cooking loss, hardness, and gumminess	[96]
	*P. sajor-caju*/fresh ground	Decreased fat content	[97]
	*P. ostreatus*/blanched	No significant differences in terms of appearance, aroma, taste, and texture in comparison with control	[98]
Beef and chicken patties	*P. sajor-caju*/fresh ground	Improved protein efficiency ratio	[99]
Pork sausages	*P. eryngii*/deep fried	Reduced fat and energy contents, while protein, moisture, total dietary fiber, cooking loss, and water holding capacity increased	[100]
Frankfurter sausages	*P. ostreatus*/flour	Fiber contents were improved as well as texture	[96]
Beef Meatballs	*P. ostreatus/* deep fried	Higher protein, fat, better protein while lower cooking losses	[101]
	*P. ostreatus*/fresh ground	It matched the physical properties to control treatment (100% meat)	[102]
	*P. ostreatus*/fresh ground	Enhanced the organoleptic characteristics and reduced the fat content	[103]
	*P. ostreatus*/fresh ground	Improved the chewiness	[104]
Salami	*P. ostreatus*/flour	Prevent lipid and protein oxidation	[105]
Beef paste	*P. ostreatus*/flour	Reduce the hardness, chewiness, and gumminess. The amino acids content and flavor component were enhanced	[106]
Abon Quail shredded meat	*P. ostreatus*/fresh ground	Reduce price and texture of quail, also increased moisture, and decreased	[107]

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
