# Peer review of "Pleurotus Genus as a Potential Ingredient for Meat Products"

_foods, 2022, doi:10.3390/foods11060779_

Round 1

Reviewer 1 Report

The article is interesting and sound. It is properly structured and based on an extensive literature search. Some minor corrections will improve the knowledge transfer.

Please discuss more the potential microbial contamination risk as well as allergies concerning the addition of Pleurotus to meat products.

Reviewer 2 Report

Foods-1613467

Type:   Review

Title:    Pleurotus genus as a potential ingredient for meat products

Mushrooms are used in most of the fields in fields (medicine, pharmacy, food, and fermentation) as they contain rich source of protein because they contain all essential amino acids, plus fibre and little fat. In fact, many countries consume mushrooms as a delicacy, particularly for their aroma and texture, fibre content, in addition to their low energy intake.

Pleurotus ostreatus is one of the most relevant edible fungi at the production level, a saprophytic fungus characterized by having bioactive compounds, mainly of phenolic origin. Therefore, the authors of this review summarized relevant studies about the composition, bioactive properties, and uses of Pleurotus spp. as a natural food additive for meat and meat products.

English is good,

Authors may have given the taxonomical classification with a neat illustration of Pleurotus ostreatus

Discussions are good, literature required is adequate

Conclusions must be elaborated since this is a review.

Accept it with minor mandatory revisions

With Regards,

Reviewer 3 Report

I have carefully revised the review by del Mar Torres-Martínez et al., entitled ‘Pleurotus genus as a potential ingredient for meat products’. The work is well organized and written (apart from few sentences to revise) and the topic is of great interest for scientists working in the field of food. I suggest to revise few points as reported below.

-I suggest to revise the introduction, since mushrooms are important source of bioactive proteins/enzymes (not only secondary metabolites) such as ribosome inactivating proteins, lectins, ribonucleases, laccases and so on. Moreover, recently a novel family of specific ribonucleases, named ‘ribotoxin-like proteins’ have been isolated from edible basidiomycetes mushrooms, including from Pleurotus ostreatus (oyster mushroom). This novel family has been retrieved only in edible mushrooms (e. g. Cyclocybe aegerita, Pleurotus ostreatus, Boletus edulis and Calocybe gambosa) having biological relevance and potential biotechnological applications in agriculture as a bio-pesticide and in biomedicine as a therapeutic and diagnostic agent. Finally, I suggest to mention also bioactive peptides from mushrooms, see for example one of the last review on updated bioactive peptides from mushrooms, published this year. Some of this information could be added also in section 4 relating to ‘bioactive compounds and their activity’ and/or in the part in which proteins/enzymes from Pleurotus are mentioned (page 2, lines 95-96, but I think that this sentence on lectins and so on could be referred to all mushrooms not only to Pleurotus, as already claimed above).

Page 2, lines 90-92: revise the sentence in order to use all plural or singular;

Page 3, line 105: don’t start a sentence with ‘because’;

Page 3, line 109: ‘Some authors suggest that…’;

Page 3, line 140: ‘important source of proteins’;

-I suggest to revise all Tables reporting only the reference number (without the name of the first author).

-Revise conclusions section… also peptides/proteins/enzymes are biologically active components of mushrooms!
